# Maternity waiting homes utilization and associated factors among childbearing women in rural settings of Finfinnee special zone, central Ethiopia: A community based cross-sectional study

**Surafel Dereje[1], Hedija Yenus[2], Getasew Amare[3], Tsegaw Amare[3]***

1 Maternal and child health office, Finfinnee special zone health department, Addis Ababa, Ethiopia,
2 Department of Reproductive Health, Institute of Public Health, College of Medicine and Health Sciences, University of Gondar, Gondar, Ethiopia, 3 Department of Health Systems and Policy, Institute of Public Health, College of Medicine and Health Sciences, University of Gondar, Gondar, Ethiopia

* tseama19@gmail.com

**Data Availability Statement:** All relevant data are within the manuscript and its Supporting Information files.

## Abstract

### Background

Maternity waiting home (MWH) is one of the strategies designed for improved access to comprehensive obstetric care for pregnant women living far from health facilities. Hence, it is vital to promote MWHs for pregnant women in Ethiopia, where most people reside in rural settings and have a high mortality rate. Therefore, this study aimed to assess MWHs utilization and associated factors among women who gave birth in the rural settings of Finfinnee special zone, central Ethiopia.

### Methods

A community-based cross-sectional study was conducted from 15th October to 20th November 2019 among women who gave birth in the last six months before data collection. Multi-stage random sampling was employed among 636 women from six rural kebeles to collect data through a face-to-face interview. Multivariable logistic regression analysis was fitted, and a 95% confidence level with a p-value <0.05 was used to determine the level and significance of the association.

### Results

Overall, MWHs utilization was 34.0% (30.3% - 37.7%). The higher age (AOR: 4.77; 95% CI: 2.76–8.24), career women (AOR: 0.39 95% CI: 0.20–0.74), non-farmer husband (AOR: 0.28; 95% CI: 0.14–0.55), rich women (AOR:1.84; 95% CI: 1.12–3.02), living greater than 60 minutes far from a health facility (AOR: 1.80; 95% CI: 1.16–2.80), and four and more live-births (AOR: 5.72; 95% CI: 1.53–21.35) significantly associated with MWHs utilization. The common services provided were latrine, bedding, and health professional checkups with

**Funding:** The author(s) received no specific funding for this work.

**Competing interests:** The authors have declared that no competing interests exist.

98.2%, 96.8%, and 75.4%, respectively. Besides, feeding service was provided by 39.8%. The primary reason not to use MWHs was the absence of enough information on MWHs.

## Conclusion

One-third of the women who delivered within the last six months utilized MWHs in the Finfinnee special zone. Our results support the primary purpose of MWHs, that women far from the health facility are more likely to utilize MWHs, but lack of adequate information is the reason not to use MWHs. Therefore, it is better to promote MWHs to fill the information gap among women with geographical barriers to reach health facilities.

## Background

Easy access to comprehensive obstetric care is a challenging issue worldwide. It is widely noted that pregnant women from the hard-to-reach areas are more likely to be exposed to obstetric complications and pregnancy-related deaths [1–3]. Therefore, the World Health Organization (WHO) introduced maternity waiting homes as one of the strategies for the safe motherhood initiative so that women have easy access to skilled obstetric care [4].

Maternity waiting homes (MWHs) are designed to help risky pregnant women and pregnant women who live far from the health facility in improving access to obstetric care after 37 completed weeks of gestation [4, 5]. Thus, the MWHs users were 80% less likely to die from pregnancy complications and 73% less likely to face stillbirth in developing countries [6].

The promotion of MWHs for countries like Ethiopia, where 80% of the total population resides in rural areas and is one of the fifteen world's "very high alert" countries according to the WHO's Fragile States Index [7], is crucial. The MWHs service was introduced in Ethiopia in the late 1980s [8] and during the time, Ethiopia was one of the five countries responsible for the world's highest maternal mortality, rate, with a maternal mortality rate of 1,061/100,000 live births [9]. According to scholars, the introduction of MWHs service contributed to the 80% reduction in maternal mortality and stillbirth in Ethiopia [6]. In addition, a meta-analysis showed that facilities having MWHs for women with a risk of pregnancy-related complications had a 47% and 49% lower risk of perinatal mortality and direct obstetric complication rate than facilities without MWHs, respectively [10].

Thus, with the growing interest in MWHs, the federal ministry of health designed a policy and strategy that promote MWHs and integrated MWHs into the health sector transformation plan to improve maternal and child health in Ethiopia [11]. However, the uptake of MWHs in Ethiopia is not in line with its expected level to achieve its goals.

Studies in Gamo Gofa [12], Jimma [13], northwest Ethiopia [14], and Benchi Maji, southern Ethiopia [15] assessed the intention of pregnant women to use MWHs. The studies showed that women's childbirth history, experience in MWH use, perceived behavioural control, having companions for facility visits, wealth status, ANC use, decision-making power for service use were determinants for intention to use MWH [12–16].

In addition, qualitative studies suggested that perceived good quality, integrated health services, awareness of pregnancy-related complications, and the husband's support in overcoming barriers were facilitators to use MWHs. On the other hand, missed work and loss of care of children at home, absences of sufficient basic facilities, poor quality, low varieties of food, and lack of entertaining services were barriers to MWH utilization [17–19].

Most of the studies focused on assessing the intention of mothers to utilize MWHs for their recent delivery. However, little is known in Ethiopia on the utilization of MWHs among actual mothers who gave birth in rural settings and their experience of the service utilization is not well explored. But, as per our knowledge, only a study conducted in Jimma, southern Ethiopia conducted on actual mothers who gave birth and which revealed that only 7% of women ever utilized MWHs [20]. However, it is not enough to explore such policy influencing intervention. Therefore, this study aimed to generate additional evidence on the uptake of MWHs among women who gave birth and factors associated with MWHs utilization in rural settings of central Ethiopia. The study will help health sector managers and policymakers to improve the uptake of MWHs services and factors that facilitate or diminish the uptake of MWHs, with the ultimate aim of achieving universal maternal health coverage.

## Materials and methods

### Study design and setting

A community-based cross-sectional study design with a quantitative method was conducted from 15th October to 20th November 2019 in the rural settings of the Finfinnee special zone. Finfinnee special zone had a total population of 649,403 in 2019, of whom, 318,207 were women, and 22,534 were pregnant [21]. The Finfinnee special zone has one administrative town, six rural districts, and 153 administrative kebeles (the smallest administrative division in Ethiopia). Based on evidence obtained from the zonal health department, approximately on average 72 childbirths were conducted over the last six months in each kebele.

According to Ethiopia's three-level healthcare delivery structure, the rural population is covered under the primary level health care delivery that includes the primary hospitals, the health centers and the health posts in which essential and non-specialized health services are provided. Out of 27 health centers in the rural kebeles of the zone, 18 of them had MWHs and were delivering free maternal health care, including health professionals' checkups, bedding, and food services. The pregnant women became aware of the services during the home visits by health extension workers, health development armies, ANC follow-up, women's conferences, and other social events [8]. In Ethiopia, the MWH service started a four-decade ago with public support. Accordingly, most MWHs services are provided without government funds free of charge [22].

### Populations

The source population for this study was all women who gave birth in the past six months of the data collection period in the six rural districts of the Finfinnee special zone. The study population was all women who gave birth in the past six months in selected rural kebeles. Those mothers who gave birth in the last six months and lived 9.5 kilometers away from health facilities were included in the study. The distance of the women's home and birth status was obtained from the health extension workers. However, women who were seriously ill during data collection time and who lived in the selected kebeles for less than six months (informal residents) were excluded from the study.

### Sample size determination and sampling procedure

A single population proportion formula was used for sample size calculation based on the assumptions for the proportion of MWHs utilization in Jimma zone of southern Ethiopia 38.7% [13], 95% confidence level, 5% margin of error, 1.5 design effect, and 5% non-response rate. Therefore, the calculated sample size was 574. The sample size for independent variables

was calculated with Epi info version 7 software with an assumption of 95% confidence level, 5% margin of error, and power of 80%. In the previous study [20], distance to a health facility was significantly associated with MWHs utilization with an adjusted odds ratio of 2.4. Thus, the estimated sample size was 636. Thus, 636 (the largest) became the final sample size required for this study.

A multistage random sampling technique was employed in the six rural districts. Eighteen out of 153 rural kebeles of the six districts that didn't have health facilities within a 9.5 km radius were eligible for the sampling. Six rural kebeles out of the 18 rural kebeles were selected with the highest population size in the first stage. In the second sampling stage, after a proportional allocation to the number of households in each kebele, all households within each kebeles were selected by systematic random sampling technique based on the order of the households on the sampling frame obtained from the health extension workers. The total sample of women delivered within the last six months of the selected kebeles was 1,282, and the sampling interval was 2. Hence, every 2nd household was visited until we got 636 selected postpartum women. When more than one eligible respondent was in the household, one respondent was randomly selected by a lottery method. A repeated visit of the women was employed when the women were absent from the home. After the three visits, the home next to the selected household was included in the study.

## Study variables

The outcome variable of this study was the utilization of maternity waiting homes defined as staying at maternity waiting homes reported by women for recent delivery/pregnancy (yes or no), which can be antenatal or postnatal. The independent variables of this study were sociodemographic characteristics of the respondents; age, religion, ethnicity, marital status, educational status, husbands' educational status, occupation, husbands' occupation, wealth index, access to transportation, and time taken to the nearest health facility. The obstetric related factors were the number of pregnancies, ANC visit for recent birth, number of ANC visits, birth preparedness plan for the recent birth, number of live births, place of the last birth, PNC follow up for recent birth, heard of MWHs, source of information, the reason to use MWHs, waiting time to get MWHs service, satisfaction with MWHs utilization, services received during the stay, reasons not to use MWHs and husband support to use MWHs. In addition, a principal component analysis was employed to create the wealth index of the women based on information on asset ownership, the number of animals owned, electricity supply to the home, health insurance, drinking water source, type of toilet, and type of materials used for construction of floors in the house. Finally, the wealth index was categorized as poor, medium, and rich. The lowest 33% of households according to the economic status variable were classified as poor; the highest 33% as rich, and the rest as average (medium) wealth index. To avoid recall bias, women who gave birth within the last six months were interviewed for their most recent delivery.

## Data collection procedures and quality control

A face-to-face interview of 30 min was employed to collect data using a pretested and structured questionnaire adapted after reviewing literature with a related topic and conceptualizing the factors significantly associated with MWHs utilization [12–15, 20, 23]. The questions were designed in such a way that the interviewer and the respondents easily understood what was intended to ask. The questionnaire was prepared in English first and then translated into Affan Oromo (the local language in the study area) then back-translated to English by language experts to check its original meaning. It consists of questions related to the sociodemographic

characteristics and obstetric characteristics, and factors related to the experience of MWHs in the pregnancy period. The data were collected by six diploma nurses and supervised by three bachelor health officers after the two days of training, mainly on the tools' contents. In addition, a pretest was conducted on 32 (5%) postpartum women at Akaki district of Finfinnee Special Zone, and necessary corrections were made on language clarity and steps of the questions before the actual data collection was conducted.

### Data management and analysis

After data collection was completed, questionnaires were checked for completeness. The completed data was coded and entered into EpiData 4.6 version software. After exporting to Stata version 14.0, incomplete, improperly formatted, duplicated, or irrelevant records were cleaned. The results of the descriptive analysis were tabulated using frequency and percent. Variables with p-value <0.2 under bivariable logistic regression were fitted for multivariable logistic regression. Adjusted odds ratio (AOR) with a 95% confidence level and a p-value less than 0.05 were used to measure the precision of the association estimate and its significance of association, respectively.

### Ethical considerations

This study was conducted following the Declaration of Helsinki. Ethical clearance was obtained from the ethical review committee of the Institute of Public Health, the University of Gondar, with the reference number IPH/676/2/2019. A supporting letter was obtained from the Finfinnee special zone health office. The study objective was explained, and both oral and written informed consent was obtained from the household head and the respondent women.

## Results

### Sociodemographic characteristics of the respondents

In this study, 636 women who gave birth in the last six months participated and 630 (99.3) were available on the random selection and 5 (0.7%) were included with a replacement for women who were absent during data collection with three repeated visits. The mean age of the respondents was 30.04 (±6.32SD) years. The majority (79.2%) of the women were housewives, and almost all (97%) were married. Besides, more than half (57.1%) were living far from the health facility, which gives maternal health services needing at least one hour of car transportation from the home of the women to reach the nearest health facility for maternal health services (Table 1).

### Obstetric characteristics of respondents

In this study, most (75.3%) of women had a history of four or fewer pregnancies. Besides, more than half (57.6%) of the respondents delivered at the health facility in the recent childbirth during the last six months. But only 216 (34.0%) women used the MWHs service. The most common (52.3%) reason not to use MWHs was lack of information on MWHs services (Table 2).

### Factors associated with MWHs utilization

The study showed that career women were 58% (AOR: 0.42; 95% CI: 0.22–0.80) less likely to use MWHs than housewives. Women whose husbands' occupations were non-farming were 82% (AOR: 0.18; 95% CI: 0.09–0.33) less likely to utilize MWHs than women with farmer

**Table 1. Sociodemographic characteristics of childbearing women in Finfinnee special zone of central Ethiopia (N = 635).**

| Characteristics | Category | Frequency (N) | Percent (%) |
| --- | --- | --- | --- |
| Age | Mean (± SD) | 30 (±6.3) | |
| | Median (± IQR) | 30 (±10) | |
| | 15–19 | 10 | 1.6 |
| | 20–24 | 144 | 22.7 |
| | 25–29 | 153 | 24.1 |
| | 30–34 | 152 | 23.9 |
| | 35–39 | 117 | 18.4 |
| | 40–45 | 59 | 9.3 |
| Religion | Orthodox | 361 | 56.9 |
| | Muslim | 79 | 12.4 |
| | Protestant | 166 | 26.1 |
| | Others* | 29 | 4.6 |
| Ethnicity | Oromo | 557 | 87.7 |
| | Gurage | 40 | 6.3 |
| | Amhara | 24 | 3.8 |
| | Others** | 14 | 2.2 |
| Marital status | Married | 616 | 97.0 |
| | Unmarried | 19 | 3.0 |
| Educational status | Not educated | 58 | 9.1 |
| | Primary level | 216 | 34.0 |
| | Secondary level | 361 | 56.9 |
| Husbands' educational status | Not educated | 91 | 14.3 |
| | Primary level | 167 | 26.3 |
| | Secondary level | 377 | 59.4 |
| Occupation | Housewife | 503 | 79.2 |
| | Others*** | 132 | 20.8 |
| Husbands' occupation | Farmer | 528 | 83.1 |
| | Others**** | 107 | 16.9 |
| Wealth index | Poor | 210 | 33.1 |
| | Medium | 213 | 33.5 |
| | Rich | 212 | 33.4 |
| Access to transportation | Easy | 277 | 43.6 |
| | Difficult | 358 | 56.4 |
| Time takes to the nearest health facility | Less than 60 minutes | 272 | 42.8 |
| | Greater than 60 minutes | 363 | 57.2 |

*Catholic and Wakefata

**Tigray and Wolayita

***Merchant, Government employee and Farmer

****Merchant, Carpenter, and Driver.

husbands. Wealthiest women were 2.51 (AOR:2.51; 95% CI: 1.57–4.01) times more likely to use MWHs than poor women. Women who were living 60 minutes far from a health facility were 1.61 (AOR: 1.61; 95% CI: 1.06–2.47) times more likely to use MWHs than women living less than 60 minutes far from health facilities. Women with four live births were 4.87 (AOR: 4.87; 95% CI: 1.38–17.17) times more likely to use MWH than women with four and fewer live births (Table 3).

**Table 2. Obstetric characteristics of childbearing women in Finfinnee special zone of central Ethiopia (N = 635).**

| Characteristics | Category | Frequency (N) | Percent (%) |
|---|---|---|---|
| Number of pregnancies | ≤4 | 478 | 75.3 |
| | >4 | 157 | 24.7 |
| ANC visit for recent birth | No | 131 | 20.6 |
| | Yes | 504 | 79.4 |
| Number of ANC visits (N = 504) | 1 | 31 | 6.1 |
| | 2 | 200 | 39.7 |
| | 3 | 88 | 17.5 |
| | ≥4 | 185 | 36.7 |
| Birth preparedness plan for the recent birth | No | 397 | 62.5 |
| | Yes | 238 | 37.5 |
| Number of live births | ≤4 | 516 | 81.3 |
| | >4 | 119 | 18.7 |
| Place of the last birth | Health facility | 366 | 57.6 |
| | Home | 269 | 42.4 |
| PNC follow up for recent birth | No | 185 | 29.1 |
| | Yes | 450 | 70.9 |
| Heard of MWHs | No | 269 | 42.4 |
| | Yes | 366 | 57.6 |
| Source of information (N = 366) | Health professional | 341 | 93.2 |
| | Others* | 25 | 6.8 |
| Used MWHs for recent birth | Not used | 419 | 66.0 |
| | Used | 216 | 34.0 |
| Reason to use MWHs | Expected complication | 6 | 2.7 |
| | To get rest | 3 | 1.4 |
| | To get better health care | 49 | 22.7 |
| | Fear of death | 135 | 62.5 |
| | To get a healthy child | 23 | 10.7 |
| Waiting time to get MWHs service (N = 216) | Less than 30 minutes | 86 | 39.8 |
| | Greater than 30 minutes | 130 | 60.2 |
| Satisfaction with MWHs utilization (N = 216) | Not Satisfactory | 50 | 23.2 |
| | Satisfactory | 166 | 76.8 |
| Services received during the stay | Latrine | 212 | 98.2 |
| | Bedding | 209 | 96.8 |
| | Health professional's check-up | 163 | 75.4 |
| | Electricity | 152 | 70.3 |
| | Meals | 86 | 39.8 |
| | Coffee | 86 | 39.8 |
| | Clean water | 67 | 31.0 |
| | Bathing | 26 | 12.0 |
| Reasons not to use MWHs | Absence of MWHs | 9 | 2.1 |
| | Absence of skilled attendant in MWHs | 8 | 1.9 |
| | Cultural influence | 8 | 1.9 |
| | Distance from home | 50 | 11.9 |
| | Lack of transportation | 14 | 3.3 |
| | Child care at home | 60 | 14.3 |
| | Lack of information | 219 | 52.3 |
| | No money | 50 | 11.9 |
| | Husband not permitted | 1 | 0.2 |

*(Continued)*

**Table 2.** (Continued)

| Characteristics | Category | Frequency (N) | Percent (%) |
|---|---|---|---|
| Husband support to use MWHs (N = 216) | No | 87 | 40.3 |
| | Yes | 129 | 59.7 |

*Peers, husband, and mass media.

## Discussion

Most preventable maternal mortalities are caused by inaccessible maternal health services or delays in providing health services [24, 25]. Hence, MWHs play a significant role in reducing maternal mortality due to preventable obstetric complications.

In this study, the magnitude of MWHs utilization is 34.0% (95% CI: 30.3% - 37.7%). The proportion of MWHs utilization in this study is higher than the magnitude in Jimma, southern Ethiopia [20], where only 7% of the women utilized MWHs on their childbirth. The difference might be attributed to sample size. The sample size in the study conducted in Jimma was six

**Table 3. Factors associated with MWHs utilization among childbearing women at Finfinnee special zone of central Ethiopia (N = 365).**

| Variable | MWHs non user | MWHs user | Proportion (%) | COR (95% CI) | AOR (95% CI) |
|---|---|---|---|---|---|
| **Total** | **N = 419 (65.98%)** | **N = 216 (34.02%)** | | | |
| Husbands' educational status | | | | | |
| Not educated | 63 | 28 | 9.1 | 1 | 1 |
| Primary level | 88 | 79 | 34.0 | 2.01 (1.18–3.46) * | 1.31 (0.67–2.60) |
| Secondary level | 268 | 109 | 56.9 | 0.91 (0.56–1.50) | 0.60 (0.32–1.14) |
| Occupation | | | | | |
| Housewife | 309 | 194 | 79.2 | 1 | 1 |
| Career woman | 110 | 22 | 20.8 | 0.31 (0.19–0.52) * | 0.42 (0.22–0.80) * |
| Husbands' occupational status | | | | | |
| Farmer | 327 | 201 | 83.1 | 1 | 1 |
| Other than farmer* | 92 | 15 | 16.9 | 0.26 (0.15–0.47) * | 0.18 (0.09–0.33) * |
| Wealth index | | | | | |
| Poor | 134 | 79 | 33.1 | 1 | 1 |
| Medium | 198 | 12 | 33.5 | 0.10 (0.05–0.19) * | 0.09 (0.05–0.19) * |
| Rich | 87 | 125 | 33.4 | 2.44 (1.64–3.60) * | 2.51 (1.57–4.01) * |
| Access to transportation | | | | | |
| Easy | 172 | 105 | 43.6 | 1 | 1 |
| Difficult | 247 | 111 | 56.4 | 0.74 (0.53–1.02) | 0.93 (0.61–1.45) |
| Time takes to the nearest health facility | | | | | |
| Less than 60 minutes | 192 | 80 | 42.8 | 1 | 1 |
| Greater than 60 minutes | 227 | 136 | 57.2 | 1.44 (1.03–2.01) * | 1.61 (1.06–2.47) * |
| Number of pregnancies | | | | | |
| ≤4 | 342 | 136 | 75.3 | 1 | 1 |
| >4 | 77 | 80 | 24.7 | 2.61 (1.80–3.78) * | 0.38 (0.10–1.31) |
| Number of live births | | | | | |
| ≤4 | 376 | 140 | 81.3 | 1 | 1 |
| >4 | 43 | 76 | 18.7 | 4.75 (3.11–7.23) * | 4.87 (1.38–17.17) * |

*Significant at P-value<0.05, COR: Crud Odds Ratio, AOR: Adjusted Odds Ratio.

times larger than the sample size used in this study. The difference might also be due to the study settings difference that central Ethiopia has more institutional delivery and more exposure to information on MWHs service than southern Ethiopia [26]. However, the result is lower than the studies conducted in Jimma [13], Benchi Maji [15], Keffa [27], Gamo Goffa [12], and East Bellesa [14], where 38.7%, 39%, 42.5%, 48.8%, and 65.3% of the pregnant women were intended to use the MWHs for their most recent delivery, respectively. This shows the different intention towards the MWHs among different settings of Ethiopia and the huge gap between actual use and the intended use of MWHs throughout the country. The variation might be due to the difference in consistent promotion of MWH services for the pregnant mother until the expected date of delivery. The difference might also be due to poor birth preparedness and complication plans among pregnant mothers.

Furthermore, this study's magnitude of MWHs utilization aligns with studies in rural Zambia [28, 29], where over a third of women utilized MWHs. However, the result is higher than the studies in rural Zambia [30], and Kenya [31], where 27.3% and 10% utilization of MWHs, respectively. The discrepancy might be due to the mobilization of health extension workers and women's health developmental army in advocating maternal health services by the government of Ethiopia [32, 33].

The most common service received by pregnant women during their stay in MWHs is latrine, and bedding, followed by a health professional check-up. However, a significant proportion of women didn't get a meal (40%) or clean water (31%) service. This indicated that despite the health professional checkup being performed well, the basic accommodation services for pregnant women are not yet fulfilled. However, among non-users, the most common reason not to use MWHs is the lack of enough information on the services provided and who is eligible to use MWHs. This finding indicates the gap in promoting MWHs services and its benefits for pregnant women. Furthermore, the qualitative study in rural Southwest Ethiopia also showed women didn't understand the aim and benefits packages of MWHs utilization [18].

The study also showed that women with more than four live births had 4.87 times higher odds of using MWH than women with four and fewer live births. This might be due to the experience as they give more birth towards more information access with their adult peers, and having a high awareness of obstetric complications. Besides, the odds of MWHs utilization among career women were 52% less likely than housewives. It might be due to housewives may take special care of themselves as they have more time than career women. This also suggests that if MWHs performed well enough in the country, institutional delivery and accessible maternal health services might be improved as more housewives reside in rural settings and have low access to a health facility. The result is in line with the study conducted in Jimma [20] but contrary to the finding from the study in Gamo Gofa, southern Ethiopia [12]. The results also supported the finding of this study that women whose husbands were non-farmer were 82% less likely to utilize MWHs than those with farmer husbands. This might be due to women getting a husband accompanied during their pregnancy and being motivated to utilize MWHs as farming is a home take job in most of rural Ethiopia. The results from the study conducted in the Jimma zone suggest women receiving accompany during their facility visit from their husbands have higher odds of using MWHs [20]. In Ethiopian settings, farmers and housewives reside in the most remote areas of the country, so this finding suggests scaling up of the MWHs service in the rural settings of the country would be helpful for getting timely obstetric care.

In this study, the odds of MWHs utilization among the wealthiest women were 2.51 times higher than poor women. It is in line with studies conducted in Jimma [20], Belessa district, northwest Ethiopia [14], Butajira hospital [34], and rural Ethiopia [35]. Even though MWHs

and other maternal health services are free of charge in Ethiopia, the low uptake among poor women might be related to inadequate exposure to MWHs service information, transport fees, and other related charges. The finding is also associated with the result that women far from a health facility have higher odds of utilizing MWHs. The studies conducted elsewhere [13,16, 20, 35–37] also indicated that women far from the health facility are more likely to utilize MWHs. This is also in line with the mission of MWHs, which mainly targeted women from the most remote areas with difficulty of transportation access and possible complication risks, accessible to maternal health services by breaching its geographic inaccessibility [38].

The study has limitations that have to be considered while interpreting and concluding the results. The study might be prone to social desirability bias as health workers were used for data collection. The study also might be prone to recall bias. But to minimize recall bias, women who gave birth within the last six months were interviewed for their most recent delivery. The sampling frame obtained from health extension workers might be misleading and outdated. The study might be prone to potential selection bias in the replacement of women with those that were available. The study's cross-sectional nature cannot establish a causal relationship between the independent and outcome variables.

## Conclusion

Overall, one-third of the postpartum women who delivered within the last six months in the Finfinnee special zone of central Ethiopia utilized MWHs. The study also indicated that the age of women, housewives, women living far from health facilities, women with non-farmer husbands, and rich wealth status contributed to utilizing MWHs. Despite latrine, bedding, and health professional checkup services commonly provided at MWHs, a significant proportion of women didn't get a meal or clean water service. Therefore, it is better to equip MWHs with basic accommodation services. Besides, the primary reason not to use MWHs among non-users was the lack of enough information on the services provided and the aim of MWHs. Therefore, it is better to promote MWHs utilization, its aim, and benefits among pregnant women through existing maternal health services like antenatal care.

## Supporting information

**S1 Data. MWHs.**
(DTA)

**S1 File. English version questionnaire.**
(DOCX)

## Acknowledgments

We want to express our thankfulness to the study participants and data collectors for their contribution to the success of this study.

## Author Contributions

**Conceptualization:** Surafel Dereje, Tsegaw Amare.

**Data curation:** Surafel Dereje.

**Formal analysis:** Getasew Amare, Tsegaw Amare.

**Methodology:** Hedija Yenus, Getasew Amare.

**Writing – original draft:** Hedija Yenus, Getasew Amare, Tsegaw Amare.

**Writing – review & editing:** Surafel Dereje, Hedija Yenus, Tsegaw Amare.

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
