## [Decision Letter · Decision Letter 0]

2 Aug 2021

PONE-D-21-18428

Maternity waiting homes utilization and associated factors among childbearing women in rural settings of Finfinnee special zone, central Ethiopia: a community based cross-sectional study

PLOS ONE

Dear Dr. Amare,

Thank you for submitting your manuscript to PLOS ONE. After careful consideration, we feel that it has merit but does not fully meet PLOS ONE’s publication criteria as it currently stands. Therefore, we invite you to submit a revised version of the manuscript that addresses the points raised during the review process.

We look forward to receiving your revised manuscript.

Kind regards,

Orvalho Augusto, MD, MPH

Academic Editor

PLOS ONE

Journal Requirements:

2. Please include additional information regarding the survey or questionnaire used in the study and ensure that you have provided sufficient details that others could replicate the analyses. For instance, if you developed a questionnaire as part of this study and it is not under a copyright more restrictive than CC-BY, please include a copy, in both the original language and English, as Supporting Information. If the original language is written in non-Latin characters, for example Amharic, Chinese, or Korean, please use a file format that ensures these characters are visible.

3. Please state whether you validated the questionnaire prior to testing on study participants. Please provide details regarding the validation group within the methods section.

Additional Editor Comments:

This report contributes to the knowledge of utilization of maternity waiting homes (MWH) in rural Ethiopia and perhaps Sub-Saharan Africa. however , there are important issues that need to be addressed:

Major issue:

- It is unclear in the whole document who should use the MWH. Therefore, it is hard to appreciate the results. The introduction is written in a way that suggests, implicitly, every woman should use the MWH service. We expected that the study setting would cover the Police or Indications for MWH in Ethiopia.

Minor issues:

1. General

- Decimal places: put 1 decimal place for proportions, means, quantiles (medians, terciles and quartiles); and 2 decimal places for standard errors and deviations, odds-ratios and their confidence intervals.

- Please have some English revision.

2. Introduction

- Please see the major issue above

- Lines 54 and 55. What is this “highest maternal mortality” compared to? Ethiopia versus other countries? Or within Ethiopia?

- Line 71 remove the “in contrast”

3. Methods

- Please see the major issue above

- Lines 97 and 98: Why just focus on within 9.5 km of a health facility. How did this decision come about?

- Sample size considerations lines 106 to 108. This statement [The sample size for factors was calculated with Epi Info version 7 software with the an assumption of 95% confidence level, 5% margin of error and power of 80%] has no sense here or it is at least incomplete. What is the magnitude of association expected to be detected?

- Statistical software: please write the names correctly. EpiData [not Epi-data] and Stata [not STATA]. Please add proper citations for the software.

4. Results:

- Please when describing the mean (m) and standard deviation (sd) do not write “m ± sd” as in lie 163. Please make m (SD: sd) or something related.

- Why are the age categories using very strange groups in table 1 and 3? The data presented in this manuscript may be used in future for meta-analysis (for example) so please use more common group divisions as multiples of 5 (< 25, 25- 34, >- 35).

- Table 1: add mean, sd, median and interquartile range of age.

- Table 1: what means “Career woman” in occupation?

- The paragraph starting at line 170 opens with the “study showed” and it is repeated on line 173.. Please avoid this. That is interpretative. It would be OK in discussion not in the results section.

- Table 2: why the 4 is used for dichotomization of previous pregnancies?

- Table 2: please sort by frequency the “services received during the stay” and the “reasons not to use MWHs”.

- Table 3: remove the “N = 419 (64.98%)” and “N = 216 (34.02%)” from the header of the table. Add one column between MWHs and COR (95% CI) for proportion [prevalence] of MWHs use. Add a row for totals [here is where this 419 and 216 will be].

- Table 3: review the stars for the significant results. For example, in the column for the COR in the second category of age the confidence interval does not include the NULL. So why no star?

Reviewers' comments:

Reviewer's Responses to Questions

**Comments to the Author**

1. Is the manuscript technically sound, and do the data support the conclusions?

Reviewer #1: No

2. Has the statistical analysis been performed appropriately and rigorously? 

Reviewer #1: No

3. Have the authors made all data underlying the findings in their manuscript fully available?

Reviewer #1: Yes

4. Is the manuscript presented in an intelligible fashion and written in standard English?

Reviewer #1: No

5. Review Comments to the Author

Reviewer #1: An interesting choice of topic. I hope you will find the comments useful in strengthening the manuscript and making it easier for the reader to judge the evidence presented.

ABSTRACT

• Line 22 – incomplete sentence, word missing

• Results: provide more specific details about how you measured (i) physical separation (“far” is too vague) and (ii) wealth (“rich” is vague). Indicate percentages of women reporting common services + indicate whether food was provided or not.

• Recommendation unclear – what do you mean scale up to women living far

BACKGROUND

General comments: (i) Existing literature on factors associated with MWH use in Ethiopia and other low-resource settings has not been adequately presented despite being available

(ii) There seems to an inherent assumption in this work that MWH stay is necessary for all pregnant women; however, they are a strategy to improve access for women who experience geographical barriers and in some cases who are expected to be at higher risk for complications. There has been no work carried out to determine what proportion of women fall into these categories in Ethiopia or indeed anywhere else, so how are the authors concluding that MWHs are under-used. Underutilization based on what? The target set in the HSTP of 80% of health centres having an MWH present is not useful here.

(iii)The literature cited regarding potential impact of MWHs has been overstated without any regard to the quality of that evidence.

Specific comments:

• Line 45 – what do you mean by equal access

• Line 47 – what do you mean the WHO “launches MWHs” – does not make sense

• Line 50 – MWHs do not provide obstetric care; they offer pregnant women a place to stay while they await birth. Obstetric care is only provided at the health facility which MWHs are associated with.

• Lines 51-52 – does not make sense. Are you trying to describe how MWHs could potentially be helpful in reducing perinatal mortality? Rephrase to make clearer.

• Line 52-53 – reference #6 had potential methodological issues that need to be considered. See https://bmcresnotes.biomedcentral.com/articles/10.1186/s13104-021-05501-2. (1) The studies included in the meta-analysis were old and generally poorly designed, they were all observational in nature and therefore have potential issues with confounding, all used health-facility based samples which reduces their generalisability to the wider population and the meta-analytic method chosen was unlikely to be appropriate for a complex intervention such as MWHs.

• Lines 54-55 is not strictly true because MWHs do not directly affect mortality or morbidity; these outcomes are highly dependent on the quality of care available at the health facility where these services are actually provided. All MWHs do is offer women an opportunity to come early, prior to the onset of complications (which usually only occur in about 20% of seemingly healthy pregnancies). You are overstating the role that MWHs play in mortality and health outcomes. Please rephrase this to more accurately reflect the role that MWHs actually play.

• Lines 57-58 – rephrase how the results from the systematic review (ref #7) and reference #8 are stated – they are currently misleading as they do not acknowledge the inherent limitations in the observational nature of these studies. They do not conclusively demonstrate the effectiveness of MWHs with respect to mortality or morbidity. Making such claims is very misleading.

• Line 56-57 – once again, the evidence is overstated using the same reference.

• Lines 63-66 – listing percentages (that are highly dependent on the setting and sample) are neither informative nor relevant to your study. What is more important is discussing what factors have been identified to be associated with intended use.

• Lines 67-70 – references 12 to 14 as you have stated in the previous paragraph relate to factors associated with intended use, so why are they listed as being related to actual use? Reference 15 describes factors related to actual use in Jimma –list those. What about evidence from qualitative work that provides insight into what might be driving use? You can’t simply exclude that because you are conducting a quantitative analysis.

METHODS

General comment: There are several fundamental elements of design that are unclear. In terms of analysis, inadequate details are provided about what the variables represent, how they were selected for the multivariable model and how the multivariable model accounted for the clustered nature of the data included.

Specific comments:

• Lines 87-89: what do you mean by 47% of women expected to deliver at a health facility? Is this a target set by the Zone? You have cited the DHS as a reference, so are you instead trying to indicate what the current level of health facility delivery is? This sentence is confusing as it seems to be mixing information about services provided at MWHs (bedding and food?) and levels of facility delivery in the area. Needs rephrasing for clarity.

• Lines 89-91 needs copy editing as it is unclear due to awkward sentence construction and grammatical issues.

• Lines 98-99: how did you establish how far women lived? Provide the exact details used to determine distances – how exactly they were measured, what type of distance, what the origin (starting point) and destinations (what health facility – nearest to women’s homes? Within their kebele of residence?) were and at what point in the study this was done. Did you obtain consent from women if you measured distances between their homes and a health facility?

• Lines 97-99: Why did you select 9.5km as part of your eligibility criteria? What evidence do you have that this criterion is relevant for your setting? Scott et al (2018) used this cut-off for their work in Zambia (https://pubmed.ncbi.nlm.nih.gov/30099401/) but had formative research to establish this as the distance they wanted to test out.

• Lines 103-105: why did you base your calculations on a study from Jimma Zone looking at intended use rather than the available study on actual use, which is your outcome of interest?

• Lines 105: Why did you select a design effect of 1.5? How do you know this is reflective of the level of correlation in outcomes in your setting? Provide some justification.

• Lines 108-109: what do you mean by the “maximum estimated sample size for the independent variable”? What independent variable?

• Lines 636: How did you arrive at a final figure of n=636 with the parameters that you provided?

• Lines 110-112: in lines 85-86 you indicated that there were n=153 kebeles within the 6 study districts, so how did you end up selecting 6 from a total of now 18 kebeles?

• Lines 112-114: Describe exactly what records are maintained by health extension workers and what information you used as your sampling frame. How up to date are these records? What is their coverage? What proportion of women living in these areas may not have used services at health posts and gone directly to hospitals or health centres (e.g., those living closer to health centres)? Ensure you acknowledge the limitations of this sampling frame in the appropriate sections.

• Lines 110-119: in what way was the distance criteria of 9.5km factored into the sampling? You seem to have selected women regardless of how far they lived. This is mismatch to what you previously described as your eligibility criteria (lines 97-99).

• Lines 113: How exactly did you conduct random sampling? Please provide brief details about what you did. You describe “ordering households” but that does not provide any information on how this was random.

• Lines 117-119: It’s unclear how you included a home next to the selected house – how could you be certain that there were eligible women living there? Also, please ensure that you acknowledge in the limitations section that this strategy potentially introduces selection bias into your sample as you are now including women who were available for interview rather than an actual random sample.

• Lines 121-122: Provide more details about whether you considered length of MWH stay and whether you made a distinction between MWH stay prior to birth (antenatal stay) and MWH stay after delivery (postnatal stay). Were these distinctions made in your outcome variable? If not, indicate that you did not and considered MWH use as any length of stay and any type of stay (antenatal/postnatal).

• Lines 123-125: Provide a list of specify sociodemographic and obstetric variables considered as independent variables as well as their operational definitions as per STROBE reporting requirements (how they are defined and how they are categorized). Also, provide some justification for the choice of these variables. Did you use existing studies from Ethiopia or other countries in sub-Saharan Africa looking at MWH use to select potential explanatory factors? If yes, provide references to these studies.

• Lines 125-128: Provide details about the methodology followed and some indication of the performance of the wealth index (truncation, clustering, correlation with asset ownership) as choice of assets influences household categorization and can potentially impact how accurately it reflects wealth. See https://pubmed.ncbi.nlm.nih.gov/28822980/. Also, the beginning of the sentence does not make sense – you can’t use PCA to “assess the wealth index”.

• Lines 128-129: Why did you opt to categorize wealth by tertile? Quartiles and quintiles are likely better at capturing subtle variation in household wealth that could potentially influence MWH use. Please provide a justification or any explanation for your choice.

• Lines 129-130: Recall bias refers to a systematic difference in the accuracy or completeness of exposure information between participants with and without the outcome of interest. Can you explain which independent variables you have included in your model are likely to have this as an issue and why this would likely differ between MWH users and non-users. Also, can you explain why you think 6 months is a more reliable recall time frame than 12 months or 3 months? How did you decide on 6 months?

• Lines 137-139:Why did you require 3 individuals to supervise 6 interviewers?

• Lines 144-145: Can you describe how you cleaned the data? Provide a brief sentence or two explaining what you did.

• Lines 146-149: Why did you rely on p-values from bivariable analyses to select variables? Can you provide justification as to why a strategy that has been described to be problematic (affects the stability of multivariable models, biases standard errors, can produce confidence intervals that are falsely narrow, can result in regression coefficients that are biased high, etc --see Ch4 in Harrell’s Regression Modelling Strategies for an in-depth explanation. This is also a useful paper: https://onlinelibrary.wiley.com/doi/full/10.1002/bimj.201700067) was selected instead of using potential explanatory factors identified in other studies?

• Lines 149-150: Confidence intervals do not “measure the degree …of association”. Please rephrase to accurately reflect what information confidence intervals provide i.e., provide insight into the precision of the association estimate.

• Provide a copy of the questionnaire used as supplementary material so that readers can see what information was collected and how.

RESULTS

• Line 162 – quoting the final number of participants is not helpful for the reader to gauge the level of response since a replacement strategy was used (lines 117-119). What you need to report is how many women were approached, how many were replaced and then the final number enrolled (n=635). 99.9% does not reflect your “participation rate” because you replaced anyone that was not available.

• Lines 164 – spelling error replace “leaving” with “living”. Please seek copy editing services to ensure that all grammatical errors and unconventional sentence construction issues are addressed.

• Lines 164-165: Specify what facility you are referring to here – nearest health facility of any level? Health centre with MWH in catchment of residence? When you describe physical separation (either as travel time or distance) you also need to specify origin and destination. See this paper that outlines problems with reporting that make it difficult to understand access issues in maternal and child health: https://journals.plos.org/plosone/article?id=10.1371/journal.pone.0184432

• Table 1 – what is a “career woman”- how was this defined? What categories of responses does it include?

• Table 1 – how was husband’s education attainment established? Did you ask women about this? It’s surprising that all women were able to answer this question.

• Table 1 – how was access to transportation measured?

• Table 1 – was any type of facility considered for the variable estimating travel time? Health post? Clinic? All health centres or health centres with MWHs? District hospital?

• Lines 174-176 – What was the question asked to women regarding reasons for non-use of MWHs? Were multiple responses possible? Also, whom did you ask the question to? All women? Women who live a specific distance from a health facility? What about women who live near a road or have easy access to transport.

• Table 1 – where was the health worker check conducted? Within the MWH or in the health centre as part of routine ANC? Provide more details about the MWH services received.

• Table 3 – it is likely that there is strong correlation between women’s age, gravidity and parity. Can you report what diagnostics you conducted to check correlation between these various variables as well as for the presence of multicollinearity in your multivariable model?

DISCUSSION:

General comment: There is a distinct failure here to acknowledge the fact that MWHs are not necessarily relevant for all women such as those with relatively good access to transport, those who live close to a health facility offering obstetric care, those who live along a road, those who live in a rural town, etc. What would be the justification of separating a pregnant woman from her household and support network for an entire week if she is able to access obstetric care when she goes into labour? What about the opportunity cost MWH stay presents – women absent from their homes and farms represents potential lost income and additional expenses incurred in organizing childcare and someone to take over domestic responsibilities. How has all this been factored into your findings?

Specific comments:

• Line 200-201: On what basis are you suggesting that use is low for this region? Your study has not established the denominator of women for whom MWHs are relevant or for whom stay would be most beneficial. There is no evidence provided of what proportion of the population experiences access barriers. Are you suggesting that 100% of women in Finfinnee should use MWHs? Direct comparison of simple percentages reported in surveys conducted in other parts of Ethiopia do not provide sufficient evidence to suggest that use is low in this setting. Please explain.

• Lines 221-223: Your conclusion that “MWHs service is being executed” is a little premature. How did you establish what sort of follow up was provided to MWH users? You have not provided any details whatsoever. Are you making this conclusion based on one question put forward to women? Did you assess what sort of follow up this was? Are you referring to “follow up by a skilled birth attendant” as judged by the use of a partograph for labour which is in the Ethiopia MWH national guidelines? What is your basis for concluding that MWHs are working as they should? Also, only 40% of users (Table 2) received meals – what did the rest do? Only 30% has access to clean water. These indicators all point to the MWH not meeting requirements which is to provide acceptable accommodation to pregnant women.

• Lines 223-224: Did you consider that perhaps lack of information among non-users was because HEWs and ANC nurses who are mainly responsible for referring pregnant women to MWHs do not discuss them with women who live close enough or do not have transport issues?

• Lines 228-232: What is more likely here is that your model has two highly correlated variables – older women are more likely to have more children. I would suggest you check to see if there is multicollinearity in the model and also justify what additional information is provided by including 3 variables that are conceptually related.

• Lines 228-245 – this entire paragraph is not very well written or thought out. It has a lot of conjecture and sweeping conclusions that are not supported by the data.

• Lines 246-257 is simply re-stating results with no additional information. Listing studies without any meaningful discussion also adds little value (lines 255-257).

• Limitations that should be discussed:

o Sampling frame constructed based on health post records which may not be up to date and could exclude a proportion of women who did not seek care at the health post either because they use health centre services directly or are unable to access any services due to distance or other barriers

o Potential selection bias in replacement of women with those that were available – you can discuss how much of an issue this might be depending on how many women you had to replace in this way

Lines 261-263 do not make sense. The outcome and dependent variable refer to the same thing, not to mention despite having said that causal relationships cannot be inferred from cross-sectional data, the entire discussion section and the conclusions does it any way.

• The conclusions are over-stated.

6. PLOS authors have the option to publish the peer review history of their article (what does this mean?). If published, this will include your full peer review and any attached files.

Reviewer #1: No

---

## [Author Response · Author response to Decision Letter 0]

17 Sep 2021

Point by point responses to editor's and reviewers' comments

Editor's comments’/suggestions.

Dear Editor, 

Thank you very much for your comments and suggestions. We tried to address your comments, suggestions and questions as follows.

1. It is unclear in the whole document who should use the MWH. Therefore, it is hard to appreciate the results. The introduction is written in a way that suggests, implicitly, every woman should use the MWH service. We expected that the study setting would cover the Police or Indications for MWH in Ethiopia.

Response: Dear editor,

We tried to revise the whole document accordingly that women who had geographical barriers, difficulty accessing transportation and had a possible risk of obstetric complications are expected to use the MWHs. Besides, there are also community health workers called health extension workers who are living nearer to the community and can access the pregnant mothers and consult and link them to the accessible health centres to use MWHs. Please the whole document of the revised manuscript.

2. Decimal places: put 1 decimal place for proportions, means, quantiles (medians, terciles and quartiles); and 2 decimal places for standard errors and deviations, odds-ratios and their confidence intervals.

Response: Dear editor, 

We again revised the whole document accordingly. Please see the whole document of the revised manuscript.

3. Please have some English revision.

Response: Dear editor, 

The whole document English write up revised accordingly by English language experts.

4. Lines 54 and 55. What is this “highest maternal mortality” compared to? Ethiopia versus other countries? Or within Ethiopia?

Response: Dear editor, 

The comparison mentioned there was maternal mortality with other countries in the world. Please see page 4 from lines 59 to 61 of the revised manuscript.

5. Line 71 remove the “in contrast”

Response: Dear editor, 

We tried to correct it as per the given suggestion. 

6. Lines 97 and 98: Why just focus on within 9.5 km of a health facility. How did this decision come about?

Response: Dear editor, 

We tied to correct it as per the suggestion. It is a typological error that women who live far (9.5km away) from health facilities were included in the study. The 9.5km was selected referencing other similar literature https://journals.plos.org/plosone/article?id=10.1371/journal.pone.0209815

Please see page 6 lines 106 to 108 of the revised manuscript.

7. Sample size considerations lines 106 to 108. This statement [The sample size for factors was calculated with Epi Info version 7 software with an assumption of 95% confidence level, 5% margin of error and power of 80%] has no sense here or it is at least incomplete. What is the magnitude of association expected to be detected?

Response: Dear editor, 

We tried to correct it as per the suggestion. Please see page 7 from lines 112 to 119 of the revised manuscript.

8. Statistical software: please write the names correctly. EpiData [not Epi-data] and Stata [not STATA]. Please add proper citations for the software.

Response: Dear editor, 

We tried to correct it as per the suggestion. Please see page 9 from lines 164 to 166 of the revised manuscript.

9. Please when describing the mean (m) and standard deviation (sd) do not write “m ± sd” as in line 163. Please make m (SD: sd) or something related.

Response: Dear editor, 

We tried to correct it as per the suggestion. Please see page 10 from lines 183 to 185 of the revised manuscript.

10. Why are the age categories using very strange groups in table 1 and 3? The data presented in this manuscript may be used in future for meta-analysis (for example) so please use more common group divisions as multiples of 5 (< 25, 25- 34, >- 35).

Response: Dear editor, 

We tried to correct it as per the suggestion. Please see page 10 table 01 of the revised manuscript.

11. Table 1: add mean, sd, median and interquartile range of age.

Response: - Dear editor, 

We tried to correct it as per the suggestion. Please see page 10 table 01 of the revised manuscript.

12. Table 1: what means “Career woman” in occupation?

Response: Dear editor, 

It was to mean other than housewife woman.

13. The paragraph starting at line 170 opens with the “study showed” and it is repeated on line 173. Please avoid this. That is interpretative. It would be OK in discussion not in the results section.

Response: Dear editor, 

We tried to correct it as per the suggestion. 

14. Table 2: why the 4 is used for dichotomization of previous pregnancies?

Response: Dear editor, 

Ethiopian the average fertility rate is 4, and that is why used 4 to dichotomize previous pregnancies.

15. Table 2: please sort by frequency the “services received during the stay” and the “reasons not to use MWHs”.

Response: Dear editor, 

We tied to correct it as per the suggestion. Please see page 10 table 2 of the revised manuscript.

16. Table 3: remove the “N = 419 (64.98%)” and “N = 216 (34.02%)” from the header of the table. Add one column between MWHs and COR (95% CI) for proportion [prevalence] of MWHs use. Add a row for totals [here is where this 419 and 216 will be].

Response: Dear editor, 

We tied to correct it as per the suggestion. 

17. Table 3: review the stars for the significant results. For example, in the column for the COR in the second category of age, the confidence interval does not include the NULL. So why no star?

Response: Dear editor, 

we corrected it as per the suggestion. 

Reviewer's comments/suggestions.

Dear Reviewer, 

Thank you very much for your comments and questions. We tried to address your comments and questions as follows.

1. Line 22 – incomplete sentence, word missing

Response: Dear reviewer, 

We tried to correct it as per the suggestion. Please see page 2 from line 22 of the revised manuscript.

2. Results: provide more specific details about how you measured (i) physical separation (“far” is too vague) and (ii) wealth (“rich” is vague). 

Response: Dear reviewer, 

We tried to correct it as per the suggestion. “Far” means women who were living greater than 60 minutes from the health facility. Whereas “rich women” mean women with upper third-class wealth status out of the three classes in the principal component analysis. 

3. Indicate percentages of women reporting common services + indicate whether food was provided or not.

Response: Dear reviewer, 

We tried to correct it as per the suggestion. Please see page 2 from lines 37 to 40 of the revised manuscript.

4. Recommendation unclear – what do you mean to scale up to women living far

Response: Dear reviewer, 

We tried to correct it as per the suggestion. Please see page 3 from lines 45 to 46 of the revised manuscript.

5. Existing literature on factors associated with MWH use in Ethiopia and other low-resource settings has not been adequately presented despite being available

Response: Dear reviewer,

We tried to search the available literature and we included them in the introduction. Please the introduction section of the revised manuscript.

6. There seems to be an inherent assumption in this work that MWH stay is necessary for all pregnant women; however, they are a strategy to improve access for women who experience geographical barriers and in some cases who are expected to be at higher risk for complications. There has been no work carried out to determine what proportion of women fall into these categories in Ethiopia or indeed anywhere else, so how are the authors concluding that MWHs are under-used. Underutilization based on what? The target set in the HSTP of 80% of health centres having an MWH present is not useful here.

Response: Dear reviewer, 

As you mentioned the target beneficiaries for MWH utilization are pregnant women with high-risk symptoms and pregnant women with geographical inaccessibility to health facilities. For this study, the scope is focusing on those pregnant mothers who are living far apart from the health facilities (far from 9.5 km from the health facilities. So, the target is all pregnant women who live 9.5 km far from the health facility. The list of pregnant mothers with their basic demographic information was registered by health extension workers on the pregnant mother’s registration book.

7. The literature cited regarding the potential impact of MWHs has been overstated without any regard to the quality of that evidence. 

Response: Dear reviewer, 

we tried to exclude literature having recognized scientific problems. For instance, reference 6 on the original manuscript.

8. Line 45 – what do you mean by equal access. Line 47 – what do you mean the WHO “launches MWHs” – does not make sense

Response: Dear reviewer, 

We corrected them as per the suggestion. Please see page 4 from lines 51 to 52 of the revised manuscript.

9. Line 50 – MWHs do not provide obstetric care; they offer pregnant women a place to stay while they await the birth. Obstetric care is only provided at the health facility with which MWHs are associated.

Response: Dear reviewer, 

We corrected it as per the suggestion. Please see page 4 from line 54of the revised manuscript.

10. Lines 51-52 – does not make sense. Are you trying to describe how MWHs could potentially help reduce perinatal mortality? Rephrase to make it clearer.

Response: Dear reviewer, 

We corrected it as per the suggestion. Please see page 4 from lines 55 to 58 of the revised manuscript.

11. Line 52-53 – reference #6 had potential methodological issues that need to be considered. See https://bmcresnotes.biomedcentral.com/articles/10.1186/s13104-021-05501-2. 

Response: Dear reviewer, 

We corrected it as per the suggestion. 

12. Lines 54-55 is not strictly true because MWHs do not directly affect mortality or morbidity; these outcomes are highly dependent on the quality of care available at the health facility where these services are actually provided. All MWHs do is offer women an opportunity to come early, prior to the onset of complications (which usually only occur in about 20% of seemingly healthy pregnancies). You are overstating the role that MWHs play in mortality and health outcomes. Please rephrase this to more accurately reflect the role that MWHs actually play.

Response: Dear reviewer, 

We corrected it as per the suggestion. Please see page 4 from lines 59 to 61 of the revised manuscript.

13. Lines 57-58 – rephrase how the results from the systematic review (ref #7) and reference #8 are stated – they are currently misleading as they do not acknowledge the inherent limitations in the observational nature of these studies. They do not conclusively demonstrate the effectiveness of MWHs with respect to mortality or morbidity. Making such claims is very misleading.

Response: Dear reviewer, 

We corrected it as per the suggestion. Please see page 4 from lines 63 to 65 of the revised manuscript.

14. Line 56-57 – once again, the evidence is overstated using the same reference.

Response: Dear reviewer, 

We corrected as per the suggestion. Please see page 4 from lines 61 to 63 of the revised manuscript.

15. Lines 63-66 – listing percentages (that are highly dependent on the setting and sample) are neither informative nor relevant to your study. What is more important is discussing what factors have been identified to be associated with intended use.

Response: Dear reviewer, 

We corrected it as per the suggestion. Please see page 5 from lines 70 to 75 of the revised manuscript.

16. Lines 67-70 – references 12 to 14 as you have stated in the previous paragraph relate to factors associated with intended use, so why are they listed as being related to actual use? Reference 15 describes factors related to actual use in Jimma –list those. What about evidence from qualitative work that provides insight into what might be driving use? You can’t simply exclude that because you are conducting a quantitative analysis.

Response: Dear reviewer, 

We corrected it as per the suggestion. Please see page 5 from lines 76 to 80 of the revised manuscript.

17. General comment: There are several fundamental elements of design that are unclear. In terms of analysis, inadequate details are provided about what the variables represent, how they were selected for the multivariable model and how the multivariable model accounted for the clustered nature of the data included.

Response: Dear reviewer, 

we tried to address the issues you raised.

18. Specific comments: Lines 87-89: what do you mean by 47% of women expected to deliver at a health facility? Is this a target set by the Zone? You have cited the DHS as a reference, so are you instead trying to indicate what the current level of health facility delivery is? This sentence is confusing as it seems to be mixing information about services provided at MWHs (bedding and food?) and levels of facility delivery in the area. Needs rephrasing for clarity.

Response: Dear reviewer, 

We corrected it as per the suggestion. Please see page 6 from lines 96 to 98 of the revised manuscript.

19. Lines 89-91 needs copy editing as it is unclear due to awkward sentence construction and grammatical issues.

Response: Dear reviewer, 

We corrected it as per the suggestion. Please see page 6 from lines 98 to 102 of the revised manuscript. 

20. Lines 98-99: how did you establish how far women lived? Provide the exact details used to determine distances – how exactly they were measured, what type of distance, what the origin (starting point) and destinations (what health facility – nearest to women’s homes? Within their kebele of residence?) were and at what point in the study this was done. Did you obtain consent from women if you measured distances between their homes and a health facility?

Response: Dear reviewer, 

We corrected it as per the suggestion. Please see page 6 from lines 106 to 108 of the revised manuscript.

21. Lines 97-99: Why did you select 9.5km as part of your eligibility criteria? What evidence do you have that this criterion is relevant for your setting? Scott et al (2018) used this cut-off for their work in Zambia (https://pubmed.ncbi.nlm.nih.gov/30099401/) but had formative research to establish this as the distance they wanted to test out.

Response: Dear reviewer,

Thank you very the questions and the comments.

Yes, as you stated establishing the cut point for far distance from the health facility by following formative or start-up evaluation is good. But we were not doing that and we used the experience of Zambia for our context to estimate the far distance from the health facility for health facility accessibility since both of the countries (Zambia and Ethiopia) are under sub-Saharan countries with relatively similar characteristics.

22. Lines 103-105: why did you base your calculations on a study from Jimma Zone looking at intended use rather than the available study on actual use, which is your outcome of interest?

Response: Dear reviewer, 

we used the study from the intended use of Jimma to increase our sample size. By either means, the largest sample size was obtained from the sample size calculated from the independent variable. 

23. Lines 105: Why did you select a design effect of 1.5? How do you know this is reflective of the level of correlation in outcomes in your setting? Provide some justification.

Response: Dear reviewer, 

As there was a two-stage on reaching the samples, we have to use a design effect of 2 but since the final sample size is adequate, there was no significant difference using the design effect of 1.5 or 2 and based on the small number of women who gave birth six months prior, the final sample size was adequate for this study. Besides, the variability lost when we move from stage to stage is not as such significant since the population heterogeneity is not significantly affected which is assumed to overcome the variability lost by using the design effect 1.5. 

24. Lines 108-109: what do you mean by the “maximum estimated sample size for the independent variable”? What independent variable?

Response: Dear reviewer, 

We corrected it as per the suggestion. Please see page 7 from lines 117 to 119 of the revised manuscript.

25. Lines 636: How did you arrive at a final figure of n=636 with the parameters that you provided?

Response: Dear reviewer,

There was a missed information and now we corrected it as per the suggestion. 

26. Lines 110-112: in lines 85-86 you indicated that there was n=153 kebeles within the 6 study districts, so how did you end up selecting 6 from a total of now 18 kebeles?

Response: Dear reviewer,

There was a missed information and now we corrected it as per the suggestion. Please see page 7 from lines 120 to 123 of the revised manuscript.

27. Lines 112-114: Describe exactly what records are maintained by health extension workers and what information you used as your sampling frame. How up to date are these records? What is their coverage? What proportion of women living in these areas may not have used services at health posts and gone directly to hospitals or health centres (e.g., those living closer to health centres)? Ensure you acknowledge the limitations of this sampling frame in the appropriate sections.

Response: Dear reviewer, 

We corrected it as per the suggestion. We included the dependence on the sampling frame obtained from health extension workers included in the limitation of the study. Please see page 17 from lines 279 to 281 of the revised manuscript.

28. Lines 110-119: in what way was the distance criteria of 9.5km factored into the sampling? You seem to have selected women regardless of how far they lived. This is mismatch to what you previously described as your eligibility criteria (lines 97-99).

Response: Dear reviewer, 

We corrected it as per the suggestion.

29. Lines 113: How exactly did you conduct random sampling? Please provide brief details about what you did. You describe “ordering households” but that does not provide any information on how this was random.

Response: Dear reviewer, 

We corrected it as per the suggestion. Please see page 7 from lines 123 to 126 of the revised manuscript.

30. Lines 117-119: It’s unclear how you included a home next to the selected house – how could you be certain that there were eligible women living there? Also, please ensure that you acknowledge in the limitations section that this strategy potentially introduces selection bias into your sample as you are now including women who were available for interview rather than an actual random sample.

Response: Dear reviewer, 

There was a list of women who delivered in the facility on the health extension workers so depending on the list, households of the women were ordered. We tried to include all randomly selected women by checking the absent households two times to minimize the selection bias.

31. Lines 121-122: Provide more details about whether you considered length of MWH stay and whether you made a distinction between MWH stay prior to birth (antenatal stay) and MWH stay after delivery (postnatal stay). Were these distinctions made in your outcome variable? If not, indicate that you did not and considered MWH use as any length of stay and any type of stay (antenatal/postnatal).

Response: Dear reviewer, 

We corrected it as per the suggestion. Please see page 7 from lines 133 to 135 of the revised manuscript.

32. Lines 123-125: Provide a list of specify sociodemographic and obstetric variables considered as independent variables as well as their operational definitions as per STROBE reporting requirements (how they are defined and how they are categorized). Also, provide some justification for the choice of these variables. Did you use existing studies from Ethiopia or other countries in sub-Saharan Africa looking at MWH use to select potential explanatory factors? If yes, provide references to these studies.

Response: Dear reviewer, 

We corrected it as per the suggestion. Please see page 8 from lines 135 to 143 of the revised manuscript.

33. Lines 125-128: Provide details about the methodology followed and some indication of the performance of the wealth index (truncation, clustering, correlation with asset ownership) as choice of assets influences household categorization and can potentially impact how accurately it reflects wealth. See https://pubmed.ncbi.nlm.nih.gov/28822980/. Also, the beginning of the sentence does not make sense – you can’t use PCA to “assess the wealth index”.

Response: Dear reviewer, 

The factor of the PCA with the highest eigenvalue was used as the variable, which describes sufficiently the socioeconomic status of a household. The respective factor scores were categorized in terciles and used in the regression analysis. The lowest 33% of households according to the economic status variable were classified as poor, the highest 33% as rich and the rest as average economic status. To avoid a risk of clustering and truncation are more variables were added to the analysis. The number of variables used in this study were more than 30 and some of the variables were continuous that appear relevant in assessing household wealth.

34. Lines 128-129: Why did you opt to categorize wealth by tertile? Quartiles and quintiles are likely better at capturing subtle variation in household wealth that could potentially influence MWH use. Please provide a justification or any explanation for your choice.

Response: Dear reviewer, 

The categorization of the wealth status in to tertile is because as the study is conducted in rural Ethiopia it will be difficult to implement quartile and quantile categorization as the difference is small among the poorest three quintiles, as each group has a similar mean score.

35. Lines 129-130: Recall bias refers to a systematic difference in the accuracy or completeness of exposure information between participants with and without the outcome of interest. Can you explain which independent variables you have included in your model are likely to have this as an issue and why this would likely differ between MWH users and non-users. Also, can you explain why you think 6 months is a more reliable recall time frame than 12 months or 3 months? How did you decide on 6 months?

Response: Dear reviewer, 

The 12 months will be too long to memorize the experiences and their satisfaction towards MWHs services. On the other hand, if we take 3 months for this study, we may not get enough sample size as the study was conducted in the rural settings of Ethiopia. 

36. Lines 137-139: Why did you require 3 individuals to supervise 6 interviewers?

Response: Dear reviewer, 

This was with an assumption of assigning one data collector for one kebele and one supervisor to two data collectors. As the study was conducted in a rural setting this amount of data collectors and supervisors were needed.

37. Lines 144-145: Can you describe how you cleaned the data? Provide a brief sentence or two explaining what you did.

Response: Dear reviewer, 

We corrected it as per the suggestion. Please see page 9 from lines 165 to 166 of the revised manuscript.

38. Lines 146-149: Why did you rely on p-values from bivariable analyses to select variables? Can you provide justification as to why a strategy that has been described to be problematic (affects the stability of multivariable models, biases standard errors, can produce confidence intervals that are falsely narrow, can result in regression coefficients that are biased high, etc --see Ch4 in Harrell’s Regression Modelling Strategies for an in-depth explanation. This is also a useful paper: https://onlinelibrary.wiley.com/doi/full/10.1002/bimj.201700067) was selected instead of using potential explanatory factors identified in other studies

Response: Dear reviewer, 

This was because there were limited studies conducted on the actual users of MWHs in Ethiopia.

39. Lines 149-150: Confidence intervals do not “measure the degree …of association”. Please rephrase to accurately reflect what information confidence intervals provide i.e., provide insight into the precision of the association estimate.

Response: Dear reviewer, 

We corrected it as per the suggestion. Please see page 9 from lines 169 to 170 of the revised manuscript.

40. Provide a copy of the questionnaire used as supplementary material so that readers can see what information was collected and how. 

Response: Dear reviewer, 

We did it as per the suggestion.

41. Line 162 – quoting the final number of participants is not helpful for the reader to gauge the level of response since a replacement strategy was used (lines 117-119). What you need to report is how many women were approached, how many were replaced and then the final number enrolled (n=635). 99.9% does not reflect your “participation rate” because you replaced anyone that was not available.

Response: Dear reviewer, 

We corrected it as per the suggestion. Please see page 10 from lines 180 to 182 of the revised manuscript.

42. Lines 164 – spelling error replace “leaving” with “living”. Please seek copy editing services to ensure that all grammatical errors and unconventional sentence construction issues are addressed.

Response: Dear reviewer, 

We corrected it as per the suggestion. Please see page 10 from line 184 of the revised manuscript. 

43. Lines 164-165: Specify what facility you are referring to here – nearest health facility of any level? Health centre with MWH in catchment of residence? When you describe physical separation (either as travel time or distance) you also need to specify origin and destination. See this paper that outlines problems with reporting that make it difficult to understand access issues in maternal and child health: https://journals.plos.org/plosone/article?id=10.1371/journal.pone.0184432

Response: Dear reviewer, 

We corrected it as per the suggestion. Please see page 10 from line 186 of the revised manuscript.

44. Table 1 – what is a “career woman”- how was this defined? What categories of responses does it include?

Response: Dear reviewer, 

The career women in this study were defined as the non-housewife women that contain the government employees, merchants and farmer women.

45. Table 1 – how was husband’s education attainment established? Did you ask women about this? It’s surprising that all women were able to answer this question.

Response: Dear reviewer, 

It is common to talk about the husband's background in our setup.

46. Table 1 – how was access to transportation measured?

Response: Dear reviewer. 

It was measured only based on the women’s perception of access to transportation services.

47. Table 1 – was any type of facility considered for the variable estimating travel time? Health post? Clinic? All health centres or health centres with MWHs? District hospital?

Response: Dear reviewer, 

The health facility includes any level of health facility which gives maternal health services can be health post health centres, clinics and district hospitals.

48. Lines 174-176 – What was the question asked to women regarding reasons for non-use of MWHs? Were multiple responses possible? Also, whom did you ask the question to? All women? Women who live a specific distance from a health facility? What about women who live near a road or have easy access to transport. 

Response: Dear reviewer, 

After asking for the status of MWHs use if she didn’t use MWHs for the recent birth she was asked for the reason. The answers can be one or more among the alternatives and any else out of the alternatives.

49. Table 1 – where was the health worker check conducted? Within the MWH or in the health centre as part of routine ANC? Provide more details about the MWH services received.

Response: Dear reviewer, 

The health professional’s check-up was conducted in the MWH. 

50. Table 3 – it is likely that there is strong correlation between women’s age, gravidity and parity. Can you report what diagnostics you conducted to check correlation between these various variables as well as for the presence of multicollinearity in your multivariable model? 

Response: Dear reviewer,

The presence of multicollinearity was examined using the Variance Inflation Factor (VIF), and a variable having a VIF value (>10) was rejected. The mean VIF of the final model was 1.99.

51. There is a distinct failure here to acknowledge the fact that MWHs are not necessarily relevant for all women such as those with relatively good access to transport, those who live close to a health facility offering obstetric care, those who live along a road, those who live in a rural town, etc. What would be the justification of separating a pregnant woman from her household and support network for an entire week if she is able to access obstetric care when she goes into labour? What about the opportunity cost MWH stay presents – women absent from their homes and farms represents potential lost income and additional expenses incurred in organizing childcare and someone to take over domestic responsibilities. How has all this been factored into your findings?

Response: Dear reviewer, 

Access to transportation is the perceived access and might not be accessible actually. In our set-up, rural Ethiopia has more challenges to access transportation even during non-emergency conditions. Therefore, we can’t undermine the MWHs service due to the perception of the women easy to access transportation. In addition, this study tries to address women with a geographical barrier to reach health facilities which are measured with the distance more than 9.5 km far from the health facility to women’s home.

52. Line 200-201: On what basis are you suggesting that use is low for this region? Your study has not established the denominator of women for whom MWHs are relevant or for whom stay would be most beneficial. There is no evidence provided of what proportion of the population experiences access barriers. Are you suggesting that 100% of women in Finfinnee should use MWHs? Direct comparison of simple percentages reported in surveys conducted in other parts of Ethiopia do not provide sufficient evidence to suggest that use is low in this setting. Please explain.

Response: Dear reviewer, 

As we tried to indicate in the manuscript, the low uptake of MWHs in this study is as compared to the intended use of MWHs in the rest of the country. The intended use of MWHs studies was conducted among those in need of MWHs to use the service. Therefore, as compared to the intention, the uptake is low in this study. 

53. Lines 221-223: Your conclusion that “MWHs service is being executed” is a little premature. How did you establish what sort of follow up was provided to MWH users? You have not provided any details whatsoever. Are you making this conclusion based on one question put forward to women? Did you assess what sort of follow up this was? Are you referring to “follow up by a skilled birth attendant” as judged by the use of a partograph for labour which is in the Ethiopia MWH national guidelines? What is your basis for concluding that MWHs are working as they should? Also, only 40% of users (Table 2) received meals – what did the rest do? Only 30% has access to clean water. These indicators all point to the MWH not meeting requirements which is to provide acceptable accommodation to pregnant women.

Response: Dear reviewer, 

We corrected it as per the suggestion. Please see page 16 from lines 241 to 244 of the revised manuscript.

54. Lines 223-224: Did you consider that perhaps lack of information among non-users was because HEWs and ANC nurses who are mainly responsible for referring pregnant women to MWHs do not discuss them with women who live close enough or do not have transport issues?

Response: Dear reviewer, 

Yes, there was a lack of information feeding from the health care providers and there is a lack of information from the mass media as well.

55. Lines 228-232: What is more likely here is that your model has two highly correlated variables – older women are more likely to have more children. I would suggest you check to see if there is multicollinearity in the model and also justify what additional information is provided by including 3 variables that are conceptually related.

Response: Dear reviewer, 

We corrected as per the suggestion and after the careful analysis for the multicollinearity, we removed age as it shows high multicollinearity with the parity of the woman. Please see page 14 table 3 of the revised manuscript.

56. Lines 228-245 – this entire paragraph is not very well written or thought out. It has a lot of conjecture and sweeping conclusions that are not supported by the data.

Response: Dear reviewer, 

We corrected it as per the suggestion.

57. Lines 246-257 is simply re-stating results with no additional information. Listing studies without any meaningful discussion also adds little value (lines 255-257).

Response: Dear reviewer, 

We corrected it as per the suggestion. Please see pages 15 and 16 from lines 248 to 264 of the revised manuscript.

58. Sampling frame constructed based on health post records which may not be up to date and could exclude a proportion of women who did not seek care at the health post either because they use health centre services directly or are unable to access any services due to distance or other barriers. Potential selection bias in replacement of women with those that were available – you can discuss how much of an issue this might be depending on how many women you had to replace in this way

Response: Dear reviewer, 

We corrected it as per the suggestion. Please see the limitation section of the revised manuscript.

59. Lines 261-263 do not make sense. The outcome and dependent variable refer to the same thing, not to mention despite having said that causal relationships cannot be inferred from cross-sectional data, the entire discussion section and the conclusions does it any way.

Response: Dear reviewer, 

We corrected it as per the suggestion. Please see the limitation section of the revised manuscript.

60. The conclusions are over-stated.

Response: Dear reviewer, 

We corrected it as per the suggestion. Please see the conclusion section of the revised manuscript.

---

## [Decision Letter · Decision Letter 1]

22 Nov 2021

PONE-D-21-18428R1Maternity waiting homes utilization and associated factors among childbearing women in rural settings of Finfinnee special zone, central Ethiopia: a community based cross-sectional studyPLOS ONE

Dear Dr. Amare,

Thank you for submitting your manuscript to PLOS ONE. After careful consideration, we feel that it has merit but does not fully meet PLOS ONE’s publication criteria as it currently stands. Therefore, we invite you to submit a revised version of the manuscript that addresses the points raised during the review process.

We look forward to receiving your revised manuscript.

Kind regards,

Orvalho Augusto, MD, MPH

Academic Editor

PLOS ONE

Journal Requirements:

Additional Editor Comments:

There are still outstanding comments to be resolved as the reviewer raises below.

Reviewers' comments:

Reviewer's Responses to Questions

**Comments to the Author**

1. If the authors have adequately addressed your comments raised in a previous round of review and you feel that this manuscript is now acceptable for publication, you may indicate that here to bypass the “Comments to the Author” section, enter your conflict of interest statement in the “Confidential to Editor” section, and submit your "Accept" recommendation.

Reviewer #2: (No Response)

2. Is the manuscript technically sound, and do the data support the conclusions?

Reviewer #2: Yes

3. Has the statistical analysis been performed appropriately and rigorously? 

Reviewer #2: Yes

4. Have the authors made all data underlying the findings in their manuscript fully available?

Reviewer #2: Yes

5. Is the manuscript presented in an intelligible fashion and written in standard English?

Reviewer #2: No

6. Review Comments to the Author

Reviewer #2: The manuscript entitled “Maternity waiting homes utilization and associated factors among childbearing women in rural settings of Finfinnee special zone, central Ethiopia: a community based cross-sectional study” is a revised version of what was previously submitted by the authors for consideration for publication. Although the authors have made attempts to improve the manuscript, there are still some concerns, which I have indicated below

General comment

The English used in the manuscript is not appropriate for PLOS ONE.

Abstract

It is not clear what the authors are communicating here. Are MWHs meant to improve access or they are strategies for improved access? What do the authors mean by “it is vital to use MWH for Ethiopia”? Improved access to service delivery alone does not solve the problem of high maternal mortality

Background

Line 51: Could authors write world health organization appropriately?

Line 54-56: Sentence is confusing and needs to be revised

Line 60: A bracket is opened but was not closed

Line 70-71: Sentence does not make any sense

Line 71-75: Authors can highlight the determinants of intention to use MWHs without having to refer to the variable names as they appeared in the study (ies). For example, attended ANC can be replaced with something like women who attended ANC or use of ANC. Authors should revise to align with the sentence structure used.

Line 79: Where are the qualitative studies? I believe references 14-16 are about utilization and not intention to use.

Methods

Line 93-94: Authors should break the sentence and continue with a new one.

Does it mean the 9 health centers do not have MWHs?

Line 107: delete far. Whether far or near is determined by the distance from the reference point

Line 112-114: in the introduction, a utilization rate of 7% was quoted, but a 38.7% is quoted for the sample size? Why did authors choose the latter instead of the former to calculate the sample size? What is the single population formula and what is the purpose of the design effect?

What justified the calculation of two sample sizes and ultimately settling on one? What do authors mean by sample size for factors?

In the selection of health facilities, did the authors take into consideration health facilities that did not have MWHs? Obviously that has a potential to influence utilization rate

Discussion.

The is evidence that the utilization of MWHs may not be the only solution to promoting maternal health in Ethiopia. From the study, even women who used MWHs did not get the full benefits; some did not get meals and some did not get clean water. What would be the benefit of using MWH to a woman when she is not assured of receiving what she is expected to get there?

7. PLOS authors have the option to publish the peer review history of their article (what does this mean?). If published, this will include your full peer review and any attached files.

Reviewer #2: **Yes: **Dr Michael Boah

---

## [Author Response · Author response to Decision Letter 1]

24 Nov 2021

Point by point responses to reviewer’s comments

Dear Reviewer, 

Thank you very much for your comments and suggestions that help us to improve the quality of our manuscript. We tried to address your comments, suggestions and questions as follows.

1. The English used in the manuscript is not appropriate for PLOS ONE.

Response: Dear reviewer, 

We tried to correct the English language write up and the grammatical errors of the whole manuscript with a trained English language expert.

2. It is not clear what the authors are communicating here. Are MWHs meant to improve access or they are strategies for improved access? What do the authors mean by “it is vital to use MWH for Ethiopia”? Improved access to service delivery alone does not solve the problem of high maternal mortality

Response: Dear reviewer, 

Thank you for your concern. In Ethiopia, where maternal mortality is more prevalent and the most important cause is the inaccessibility of health services, strategies like MWH will be crucial. Moreover, we have corrected the sentence. Please see page 2 lines 24-25.

3. Line 51: Could authors write world health organization appropriately?

Response: Dear reviewer, 

We corrected it as per the suggestion. Please see page 4 line 52.

4. Line 54-56: Sentence is confusing and needs to be revised

Response: Dear reviewer, 

We corrected it as per the suggestion. Please see page 4 lines 55-58.

5. Line 60: A bracket is opened but was not closed

Response: Dear reviewer, 

We corrected it as per the suggestion. Please see page 4 line 60.

6. Line 70-71: Sentence does not make any sense

Response: Dear reviewer, 

We corrected it as per the suggestion. Please see page 5 lines 71-75.

7. Line 71-75: Authors can highlight the determinants of intention to use MWHs without having to refer to the variable names as they appeared in the study (ies). For example, attended ANC can be replaced with something like women who attended ANC or use of ANC. Authors should revise to align with the sentence structure used.

Response: Dear reviewer, 

We corrected it as per the suggestion. Please see page 5 lines 71-75.

8. Line 79: Where are the qualitative studies? I believe references 14-16 are about utilization and not intention to use.

Response: Dear reviewer, 

We corrected it as per the suggestion. Please see page 5 line 82.

9. Line 93-94: Authors should break the sentence and continue with a new one.

 Response: Dear reviewer, 

We corrected it as per the suggestion. Please see page 6 lines 93-94.

10. Does it mean the 9 health centers do not have MWHs?

Response: Dear reviewer, 

Yes, the rest (9) of the health centres have no MWHs.

11. Line 107: delete far. Whether far or near is determined by the distance from the reference point

Response: Dear reviewer, 

We corrected it as per the suggestion. Please see page 6 line 107.

12. Line 112-114: in the introduction, a utilization rate of 7% was quoted, but a 38.7% is quoted for the sample size? Why did authors choose the latter instead of the former to calculate the sample size? 

Response: Dear reviewer, 

Using the single proportion formula we calculated the final sample size separately and we got 158 employing 7% proportion of MWHs use which is calculated from only three districts of Jimma and we got 574 using 38.7% proportion of MWHs intended use. To get a more precise estimate, we used the latter one which gives the largest sample size. However, we also calculated the sample size for independent variables using the double population proportion formula and got 636 sample sizes. Hence, we used the largest sample size (636) as the final sample size of this study. 

13. What is the single population formula and what is the purpose of the design effect? 

Response: Dear reviewer, 

A single population proportion formula is a formula used to calculate the sample size needed to estimate the proportion or percentage of an outcome of interest in a population from data of which the outcome consists of two categories (dichotomous). Based on it, we employed here the single proportion formula (n = Zα/22 *p*(1-p) / d2) to estimate the sample size used to determine the proportion MWH utilization. 

A design effect is a strategy used to adjust the sampling effect by increasing the sample size when employing other than simple random sampling (cluster sampling, systematic sampling, stratified sampling, multistage sampling). Hence, we employed here multistage sampling followed by systematic sampling. Therefore, to adjust for the sampling deviation from simple random we used a design effect of 1.5. 

14. What justified the calculation of two sample sizes and ultimately settling on one? What do authors mean by sample size for factors?

Response: Dear reviewer, 

The phrase “sample size for factors” was to mean that a sample size needs to be taken to measure the association of a single independent variable with the outcome variable. Therefore, calculating the sample size both for the outcome variable and the independent variables help to reach the largest sample size which is considered to include the other small sample sizes. Moreover, we corrected the phrase. Please take a look at page 7 line 116.

15. In the selection of health facilities, did the authors take into consideration health facilities that did not have MWHs? Obviously, that has a potential to influence utilization rate

Response: Dear reviewer, 

Thank you for your very important point. We didn’t consider that because the absence of MWHs in the nearest health facility can be one of the reasons for not utilizing the MWHs. As we indicated in table 2, nine women responded that the absence of MWHs is one of the reasons not to use MWHs among women who didn’t utilize MWHs for their most recent birth. 

16. There is evidence that the utilization of MWHs may not be the only solution to promoting maternal health in Ethiopia. From the study, even women who used MWHs did not get the full benefits; some did not get meals and some did not get clean water. What would be the benefit of using MWH to a woman when she is not assured of receiving what she is expected to get there?

Response: Dear reviewer, 

Thank you. We believe that even if the services provided by the MWH is not satisfactory enough as intended by its objective, there are promising indicators that promote the utilization of MWH. Like for example in this study, 75.4% of women get health professionals' check-ups in the MWH. Therefore, even though MWHs utilisation, is not the only strategy for promoting maternal health, it will be highly important in reducing maternal mortality in risky and rural women with the difficulty of access to transportation.

---

## [Decision Letter · Decision Letter 2]

25 Jan 2022

PONE-D-21-18428R2Maternity waiting homes utilization and associated factors among childbearing women in rural settings of Finfinnee special zone, central Ethiopia: a community based cross-sectional studyPLOS ONE

Dear Dr. Amare,

Thank you for submitting your manuscript to PLOS ONE. After careful consideration, we feel that it has merit but does not fully meet PLOS ONE’s publication criteria as it currently stands. Therefore, we invite you to submit a revised version of the manuscript that addresses the points raised during the review process.

We look forward to receiving your revised manuscript.

Kind regards,

Orvalho Augusto, MD, MPH

Academic Editor

PLOS ONE

Additional Editor Comments:

We are going now to a third revision and this manuscript and we still have so many English typos. Please a native English speaker to address this.

Reviewers' comments:

Reviewer's Responses to Questions

**Comments to the Author**

1. If the authors have adequately addressed your comments raised in a previous round of review and you feel that this manuscript is now acceptable for publication, you may indicate that here to bypass the “Comments to the Author” section, enter your conflict of interest statement in the “Confidential to Editor” section, and submit your "Accept" recommendation.

Reviewer #3: (No Response)

2. Is the manuscript technically sound, and do the data support the conclusions?

Reviewer #3: Partly

3. Has the statistical analysis been performed appropriately and rigorously? 

Reviewer #3: Yes

4. Have the authors made all data underlying the findings in their manuscript fully available?

Reviewer #3: Yes

5. Is the manuscript presented in an intelligible fashion and written in standard English?

Reviewer #3: Yes

6. Review Comments to the Author

Reviewer #3: Thank you for the opportunity to review your article. This is my first review and I noted the comments of a previous review.

The article reads easily and the maternal mortality in Ethiopia is really a concern. Thank you for attending to the language. There are still many errors throughout and I have pointed out some. Using a native English speaking editor is always best prior to submitting articles to international journals. The following aspects need to be attended to:

Background

L55-L57 Edit the language.

L57 - What does final week of pregnancy mean in this study as a term pregnancy is widely accepted as 37 completed weeks?

L60 - Remove the additional bracket.

L61 - What is the ideal maternal mortality? Place Ethiopia in the international perspective.

L61-L63 - What was the maternal mortality prior to implementation of the MWHs?

L71-L80 - Many studies have been conducted on MWHs and the facilitators and barriers to use of the MWHs are known. The significance of the research problem is stated in L81-L88, but is not clear.

Study design and setting

L92 - What is a community-based cross-sectional study? The study design is not described and applied to this study.

L93-L94 - The population was 649 403. Indicate the specific year.

L97 - Approximately how many births are conducted in each kebele and what type of healthcare is present?

L107 - Why should the mothers live 9.5km away?

L119 - What is AOR 2.4?

L128 - Correct the language in the sentence 'The sampling interval...'

Study variables

L140 - There is a word missing in the sentence 'The obstetrical...'

L140 - L145 - How were the factors selected in the questionnaire? Describing the construction of the research tool is very important as it affect the validity and reliability of the study.

The results are well presented and clear.

Obstetric characteristics of respondents

L196-L202 - There is an overuse of adverts. Please remove some.

Discussion

I am concerned about the depth of the discussion. A discussion needs to be aligned with your results. I cannot differentiate between the sociodemographic characteristics, obstetric characteristics and factors associated with MWH utilization. The role of the healthcare provider is still not clear as well as the benefits of use of the MWH for the women. Perhaps create three headings in the discussion section and arrange the content below each.

L224-L225 - Correct the language in the sentence starting with 'The study showed...'

L225-L226 - The actual use in this study is compared to the intended use in other studies. There must be similar data available.

L227-L228 - Correct the language in the sentence starting with 'The studies in Benchi Maji...'

L232-L234 - Correct the language

L234-L235 - Correct the language. The sentence does not flow well.

L258 - Correct the language.

L256-L266 Correct the language of the sentence 'This might be the effect...'

L282 - Add 's' to the word 'result'.

L293 - 'Housewife women' is not appropriate English.

Acknowledgements

L302 - The word 'admire' is not appropriate English.

7. PLOS authors have the option to publish the peer review history of their article (what does this mean?). If published, this will include your full peer review and any attached files.

Reviewer #3: No

---

## [Author Response · Author response to Decision Letter 2]

2 Feb 2022

Point by point responses to reviewer’s comments

Dear Reviewer, 

We thank you for your comments, suggestions and questions that would be helpful in improving the quality of our manuscript. We tried to address all of your concerns as follows.

1. L55-L57 Edit the language.

Response: Dear reviewer, 

We have edited the language construction. Please see page 4 line 53-55.

2. L57 - What does final week of pregnancy mean in this study as a term pregnancy is widely accepted as 37 completed weeks?

Response: Dear reviewer, 

We have corrected it as per the suggestion. Please see page 4 line 54-55.

3. L60 - Remove the additional bracket.

Response: Dear reviewer, 

We have corrected it as per the suggestion. Please see page 4 line 57-59.

4. L61 - What is the ideal maternal mortality? Place Ethiopia in the international perspective.

Response: Dear reviewer, 

We have corrected the sentence. Please see page 4 line 57-59.

5. L61-L63 - What was the maternal mortality prior to implementation of the MWHs?

Response: Dear reviewer, 

We have included the maternal mortality prior to implementation of the MWHs in the revised manuscript. Please see page 4 line 61.

6. L71-L80 - Many studies have been conducted on MWHs and the facilitators and barriers to use of the MWHs are known. The significance of the research problem is stated in L81-L88, but is not clear.

Response: Dear reviewer, 

We have made clear the significance of the study. Please see page 4 and 5 from line 81- 91.

7. L92 - What is a community-based cross-sectional study? The study design is not described and applied to this study.

Response: Dear reviewer, 

A community-based cross-sectional study design is one of the study designs, which can be descriptive or analytical that involves collecting data from a population from the community at one specific point in time. In this study, we have collected data from 15th October to 20th November 2019 among reproductive women who gave birth in the past six months of delivery before the actual data collection period in the rural settings of the Finfinnee special zone. To make more clarity, we have corrected the sentence. Please see page 5 line 94.

8. L93-L94 - The population was 649,403. Indicate the specific year.

Response: Dear reviewer, 

We have indicated the year as per the suggestion. Please see page 5 line 95.

9. L97 - Approximately how many births are conducted in each kebele and what type of healthcare is present?

Response: Dear reviewer, 

We have addressed you question. Please see page 5 line 99.

10. L107 - Why should the mothers live 9.5km away?

Response: Dear reviewer, 

As we have indicated in the background, most Ethiopians live in rural settings where people are living far from health facility and we have used the cut-off point as referenced in another related study to operationalize how far is the pregnant women resides from the health facility. It has to be noted that MWHs service is mainly targeted to remote-residing pregnant women to break the geographical inaccessibility of obstetric care to reduce pregnancy complications and minimize perinatal mortality.

11. L119 - What is AOR 2.4?

Response: Dear reviewer, 

It is to mean adjusted odds ratio. We have corrected it as per the suggestion. Please see page 7 line 126.

12. L128 - Correct the language in the sentence 'The sampling interval...'

Response: Dear reviewer, 

We have edited the language construction of the sentence. Please see page 8 line 136-137.

13. L140 - There is a word missing in the sentence 'The obstetrical...'

Response: Dear reviewer, 

We have corrected it as per the suggestion. Please see page 8 line 148.

14. L140 - L145 - How were the factors selected in the questionnaire? Describing the construction of the research tool is very important as it affect the validity and reliability of the study.

Response: Dear reviewer, 

Thank you very much for your important point. However, in the “Data collection procedures and quality control” section we have detailed the description of the questionnaire development process and the data collection procedure in addition to the data quality assurance.

15. The results are well presented and clear.

Response: Dear reviewer, 

Thank you very much for your appreciation.

16. L196-L202 - There is an overuse of adverts. Please remove some.

Response: Dear reviewer, 

We have corrected it as per the suggestion. Please see page 13 line 206-209.

17. The role of the healthcare provider is still not clear as well as the benefits of use of the MWH for the women. 

Response: Dear reviewer, 

As we have indicated in the background section in detail, the benefit of the MWH is to facilitate easy access of pregnant women to obstetric care through making them near to the health facility. The service mainly targets risky pregnant women and pregnant women who live at very remote distances from health facilities. In this regard, the health facilities are expected to construct the MWHs and the health professionals should check them on a regular basis.

18. Perhaps create three headings in the discussion section

Response: Dear reviewer, 

Thank you very much for your concern about the discussion and for the suggestions on the discussion sections. We have addressed the issues you raised in the discussion section, but we haven’t created the heading under the discussion section. However, each paragraph was written assuming to contain a specific idea. For example, the first paragraph is an introduction to the discussion section. The second paragraph is about the comparison of this study’s magnitude of MWH with the other local studies. The third paragraph is about comparison with other countries findings. The fourth paragraph is about services received by pregnant women. The fifth, sixth and seventh paragraph are about the factors associated with MWHs utilization. The last paragraph stated the limitations of the study. This form of discussion section writing is common in other studies including studies which were published in PLOS ONE journal.

19. L224-L225 - Correct the language in the sentence starting with 'The study showed...'

Response: Dear reviewer, 

We have edited the language construction of the sentence. Please see page 16 line 231.

20. L225-L226 - The actual use in this study is compared to the intended use in other studies. There must be similar data available.

Response: Dear reviewer, 

Thank you very much for your concern. However, we have used the available studies in Ethiopia, which were conducted on actual users of MWH, for comparison. But due to the limited number of studies on actual users, we have also compared our findings with intended users of MWH in other settings. Even though it is not the right way of comparison, we believe that there might not be huge difference among the studies settings with our study setting since all them were conducted in Ethiopia. In addition, comparison of the actual use of MWH with the intended use of MWH might also generate another evidence which will be used for policy influence.

21. L227-L228 - Correct the language in the sentence starting with 'The studies in Benchi Maji...'

Response: Dear reviewer, 

We have corrected it as per the suggestion. Please see page 16 line 237-240.

22. L232-L234 - Correct the language

Response: Dear reviewer, 

We have corrected it as per the suggestion. Please see page 16 line 233

23. L234-L235 - Correct the language. The sentence does not flow well.

Response: Dear reviewer, 

We have corrected it as per the suggestion. Please see page 16 line 233-235.

24. L258 - Correct the language.

Response: Dear reviewer, 

We have corrected it as per the suggestion. Please see page 17 line 268.

25. L256-L266 Correct the language of the sentence 'This might be the effect...'

Response: Dear reviewer, 

We have corrected it as per the suggestion. Please see page 18 line 272-274.

26. L282 - Add 's' to the word 'result'.

Response: Dear reviewer, 

We have added it as per the suggestion. Please see page 18 line 291.

27. L293 - 'Housewife women' is not appropriate English.

Response: Dear reviewer, 

We have corrected it as per the suggestion. Please see page 19 line 301.

28. L302 - The word 'admire' is not appropriate English.

Response: Dear reviewer, 

We have corrected it as per the suggestion. Please see page 19 line 310.

---

## [Decision Letter · Decision Letter 3]

28 Feb 2022

Maternity waiting homes utilization and associated factors among childbearing women in rural settings of Finfinnee special zone, central Ethiopia: a community based cross-sectional study

PONE-D-21-18428R3

Dear Dr. Amare,

We’re pleased to inform you that your manuscript has been judged scientifically suitable for publication and will be formally accepted for publication once it meets all outstanding technical requirements.

Kind regards,

Orvalho Augusto, MD, MPH

Academic Editor

PLOS ONE

Additional Editor Comments (optional):

Reviewers' comments:

Reviewer's Responses to Questions

**Comments to the Author**

1. If the authors have adequately addressed your comments raised in a previous round of review and you feel that this manuscript is now acceptable for publication, you may indicate that here to bypass the “Comments to the Author” section, enter your conflict of interest statement in the “Confidential to Editor” section, and submit your "Accept" recommendation.

Reviewer #3: (No Response)

2. Is the manuscript technically sound, and do the data support the conclusions?

Reviewer #3: Yes

3. Has the statistical analysis been performed appropriately and rigorously? 

Reviewer #3: Yes

4. Have the authors made all data underlying the findings in their manuscript fully available?

Reviewer #3: Yes

5. Is the manuscript presented in an intelligible fashion and written in standard English?

Reviewer #3: Yes

6. Review Comments to the Author

Reviewer #3: Thank you for the much improved article. The discussion section now focuses on utilization of maternity waiting homes. There are however aspects that have not been addressed. In addition, altering the text brought forth new linguistic errors.

- L114 - The distance of 9.5 km from the maternity home is sill not clear.

- L161 - The quality control of data collection was addressed. The validity and reliability of the new questionnaire is not described. Validity includes internal and external validity and one expects face and content validity, construct validity and criterion validity. Were experts used during development of the questionnaire?

- L178 - How many records could not be used?

- L262 - Change to 'fewer than four' women.

Below some of the language errors:

- L38 - include 'are' before 'significantly'

- L39-40 - Rephrase the sentence

- L84-86 - Rephrase the sentence

- L90 - Insert 'determine' before 'factors'

- L121 - Move [23] in the middle of the sentence towards the end of the sentence

- L233 - Insert 'during' before 'their childbirth'

- L239 - Remove 'were' after 'pregnant women'

- L248 - 'utilization' should be 'utilized'

- L254 - Remove 'service'

- L254 - 'Did not' instead of 'didn't' (Correct all)

- L262 - Sentence not clear - rephrase

- L271 - Change 'farmer' to 'farmers'

- L272-276 - Rephrase poor language

- L313 - Include 'was' after 'SD'

7. PLOS authors have the option to publish the peer review history of their article (what does this mean?). If published, this will include your full peer review and any attached files.

Reviewer #3: No

---

## [Editor Report · Acceptance letter]

4 Mar 2022

PONE-D-21-18428R3 

Maternity waiting homes utilization and associated factors among childbearing women in rural settings of Finfinnee special zone, central Ethiopia: a community based cross-sectional study 

Dear Dr. Amare:

I'm pleased to inform you that your manuscript has been deemed suitable for publication in PLOS ONE. Congratulations! Your manuscript is now with our production department. 

Kind regards, 

on behalf of

Dr. Orvalho Augusto 

Academic Editor

PLOS ONE